# Joint Learning of Hierarchical Word Embeddings from a Corpus and a Taxonomy

**Mohammed Alsuhaibani**                    M.A.ALSUHAIBANI@LIVERPOOL.AC.UK
*Department of Computer Science,*
*University of Liverpool,*
*Liverpool, L69 3BX, UK*
*Department of Computer Science,*                    M.SUHIBANI@QU.EDU.SA
*Qassim University,*
*Qassim, Saudi Arabia*

**Takanori Maehara**                    TAKANORI.MAEHARA@RIKEN.JP
*RIKEN Center for Advanced*
*Intelligence Project,*
*Tokyo, 103-0027, Japan*

**Danushka Bollegala**                    DANUSHKA@LIVERPOOL.AC.UK
*Department of Computer Science,*
*University of Liverpool,*
*Liverpool, L69 3BX, UK*

## Abstract

Identifying the hypernym relations that hold between words is a fundamental task in NLP. Word embedding methods have recently shown some capability to encode hypernymy. However, such methods tend not to explicitly encode the hypernym hierarchy that exists between words. In this paper, we propose a method to learn a hierarchical word embedding in a specific order to capture the hypernymy. To learn the word embeddings, the proposed method considers not only the hypernym relations that exists between words on a taxonomy, but also their contextual information in a large text corpus. The experimental results on a supervised hypernymy detection and a newly-proposed hierarchical path completion tasks show the ability of the proposed method to encode the hierarchy. Moreover, the proposed method outperforms previously proposed methods for learning word and hypernym-specific word embeddings on multiple benchmarks.

## 1. Introduction

Hypernymy relation is an essential component for many Natural Language Processing (NLP) tasks. It represents an asymmetric relation between name of a class *(hypernym)* and a particular instance of it *(hyponym)*. For example, given the hypernymy pair *(bird, vertebrate)*, the hyponym word *bird* is a particular instance of the hypernym word *vertebrate*, and one can simply say *"a bird **is a** vertebrate"*. Hypernymy identification plays a major role in various NLP tasks such as question answering [Huang et al., 2008], taxonomy construction [Navigli et al., 2011], textual entailment [Dagan et al., 2013] and text generation [Biran and McKeown, 2013], to name a few.

The task of identifying hypernymy has been long addressed by relying on either the lexical-syntactic patterns where a particular textual pattern suggests the existence of a hypernymy relation, or the distributional representation of a given pair of words. In a pattern-based approach, from a sentence like *"a bird such as a falcon"* we can identify a hypernymy relation between *bird* and *fal-*

*con*. Despite its simplicity and efficiency, pattern-based approaches suffer from a low precision and coverage issue [Yu et al., 2015]. For example, let us consider the sentence *"some birds recorded in Africa such as Gadwall"*, a typical lexical-pattern based approach may incorrectly detect *(Gadwall, Africa)* as hypernymy.

In contrast to the requirement of the pattern-based approaches to have the hypernym and hyponym words occurring in the same sentence, the distributional family of approaches rely on the words co-occurrence statistics in a large text corpus to represent the words and then deduce the hold of hypernymy. It works on the assumption that taxonomically related words tend to occur in a similar context. In particular, a hypernym word has a broader context than its hypernym, and therefore the contextual features of the hyponym word are usually a subset of that of its hypernym. However, such approaches commonly struggle from discriminating the hypernym relations from other lexico-semantic relations [Shwartz et al., 2016b].

Lately, distributed word representations (a.k.a word embeddings) [Mikolov et al., 2013, Pennington et al., 2014] have shown some capability to encode hypernymy. Such methods typically embed the words into dense, low-dimensional, real-valued vectors, using the co-occurrence statistics obtained from a text corpora, and then used in a supervised settings to detect hypernymy. However, typical word embeddings models rely only on co-occurrence based similarity, and therefore are insufficient to encode taxonomic relations in the learnt word embeddings [Anh et al., 2016].

Several recent studies have proposed methods for learning hypernymy-specific word embeddings [Yu et al., 2015, Anh et al., 2016, Nguyen et al., 2017]. Yu et al. [2015] design a neural network to learn word embeddings that perceive hypernymy, purely relying on extracted pairwise training data from a web corpus. Moreover, Anh et al. [2016] proposed a similar approach, but further utilise the contextual information of the pre-extracted pairwise hypernymy. Similarly, Nguyen et al. [2017] introduce the HyperVec model that use the Skip-gram with Negative Sampling (SGNS) [Mikolov et al., 2013] objective to learn the word embeddings from a corpus subject to pairwise hypernymy constraints extracted from a taxonomy. Moreover, Nickel and Kiela [2017] proposed a new model that embed symbolic data into hyperbolic space, particularly into Poincaré ball, to learn hierarchical embeddings. Another model that is recently proposed is the LEAR model of Vulić and Mrkšić [2018]. LEAR is a post-processing model that takes any word vectors and then adjust the input vectors to emphasis the hypernym relations. The aforementioned methods merely consider pairwise hypernymy relations rather than the hierarchical hypernymy path connecting a word to the root in a taxonomy.

In this paper, we propose a method that jointly learns hierarchical word embeddings (HWE) from a corpus and a taxonomy. The proposed method begins by embedding the words into random low-dimensional real-valued vectors, and subsequently updates the embeddings to encode the hierarchical structure available in the taxonomy. To train the proposed method, we use a taxonomy to extract the hierarchical hypernymy paths for the hyponym words, and use the global vector [Pennington et al., 2014] as a context-aware objective between the hypernym and hyponym words. As such, the proposed model benefits from both the contextual information as well as the taxonomic relations to learn the embeddings.

In our experiments, we evaluate our hierarchical word embeddings on the standard supervised hypernymy task on various benchmarks, graded lexical entailment prediction and a newly-proposed hierarchical path completion task. On the supervised hypernymy identification, the proposed method reports an improvement over several previously proposed methods, showing the benefit of considering the full hierarchical hypernymy paths instead of only pairwise relations. Moreover, a quantitative

and qualitative analysis on the newly-proposed hierarchical path completion task further illustrate the ability of the proposed method to encode hypernymy in the learn embeddings.

## 2. Related Work

Over the years, two lines of approaches, pattern-based and distributional, have been predominantly followed in the NLP community for identifying the hypernymy relation between words. The pattern-based approach uses large text corpora to capture lexical-syntactic patterns with a high probability of representing a hypernymy relationship [Liu and Singh, 2004, Suchanek et al., 2007, Wu et al., 2012]. Popularly known as the *Hearst's patterns* [Hearst, 1992] are an example of such patterns, for instance, the sentence *"vertebrate such as birds"* indicates that *vertebrate* is a hypernym of *birds*. Extracting such patterns paves the way to automatically build a taxonomy of hypernymy relations, where the task of identifying the existence of hypernymy relation between a given pair of words is then transformed into a problem of checking their existence in such taxonomies. Unfortunately, pattern-based approaches are prone to data sparseness issues because two words must co-occur in the same context in order for a pattern to be extracted [Yu et al., 2015, Shwartz et al., 2016b].

Distributional methods, on the other hand, focus on the word distribution in text corpora to represent the words in a high dimensional vector space [Turney and Pantel, 2010], and rely on the Distributional Inclusion Hypothesis (DIH) [Geffet and Dagan, 2005] to identify the hypernymy relationships. DIH works on the assumption that a significant number of the distributional context features of the hyponym word are expected to be a subset of the hypernym word features. For instance, given the pair *(bird, vertebrate)*, DIH assumes that most of the distributional features of *bird* are features of *vertebrate* as well, but does not necessarily true for the reverse. Such unsupervised distributional methods for hypernymy identification include [Weeds et al., 2004, Clarke, 2009, Kotlerman et al., 2010, Lenci and Benotto, 2012].

Pattern-based and distribution-based approaches have complementary strengths, hence some hybrid approaches have been proposed combining the strengths of the two approaches [Mirkin et al., 2006, Shwartz et al., 2016a].

Several methods have been proposed that use pre-trained word embeddings as the input and learn a classifier that can predict whether a hypernym relation exist between a given pair of words. These methods can be seen as a combination between unsupervised and supervised methods in the sense that the pre-trained word emebddings used as the input can be learnt using unsupervised word embedding learning methods [Mikolov et al., 2013, Pennington et al., 2014], and the classifier training part can be seen as a supervised (binary) classification task, where the labelled instances for training can be retrieved from a taxonomy containing hypernyms-hyponyms. An important research question in such approaches is *how to represent the relationship between two words using their word embeddings?* To address this point, several unsupervised operators that take in two vectors (corresponding to the embeddings of the two input words) and return a vector (corresponding to the relation between the two input words) have been studied in prior work such as concatenation [Baroni and Lenci, 2011], difference [Roller et al., 2014], addition [Weeds et al., 2014] and hybrid combinations [Turney and Mohammad, 2015]. Although numerous work have reported good hypernym detection accuracies using such supervised methods [Roller et al., 2014, Weeds et al., 2014, Kruszewski et al., 2015, Sanchez and Riedel, 2017, Vulić et al., 2017]. [Levy et al., 2015b] raise doubts as to whether these methods are truly learning the hypernymy relation or simply memorising prototypical hypernyms.

More recently, a new line of work focusing on learning hypernymy-specific word embeddings [Yu et al., 2015, Anh et al., 2016, Nickel and Kiela, 2017, Glavaš and Ponzetto, 2017, Nguyen et al., 2017, Vulić and Mrkšić, 2018] has gained much popularity in the field. This line is the closest to our work. Yu et al. [2015] proposed a supervised distance-margin neural network for the purpose of encoding hypernymy properties in the learnt embeddings. The neural network is trained using only the pre-extracted pairwise hypernymy tuples from Probase [Wu et al., 2012], without considering any contextual information that might be available in sentences where the two words co-occur. However, it has been shown that co-occurrence contexts provide useful semantic clues for detecting taxonomic relationships [Velardi et al., 2013, Anh et al., 2014].

Similarly, Anh et al. [2016] proposed a dynamically weighted neural network for leaning hypernymy specific word embeddings. The proposed method initialises by extracting pairwise hypernymy data from WordNet [Miller, 1995], followed by extracting related sentences from Wikipedia to form the training data as triples including the hypernym, hyponym and a context word between them, as an input to the neural network. Nguyen et al. [2017] introduced unsupervised neural model (HyperVec) that outperformed the previous two hypernym-specific embeddings work. HyperVec jointly learns from the extracted hypernym pairs and contextual data. In particular, their proposed method starts by extracting all the hypernym pairs from the WordNet, and use SGNS as a context-aware objective and jointly learns the hypernymy-specific word embeddings. Nguyen et al. [2017] report an improvement over Yu et al. [2015] and Anh et al. [2016] methods on hypernymy relation identification task as well as the task of distinguishing between hypernym and hyponyms in a hypernymy relation pair. Moreover, Vulić and Mrkšić [2018] introduce Lexical Entailment Attract Repel (LEAR) model to learn word embeddings that encode hypernymy. LEAR works as a post-processing model that can take any word vectors as an input and inject external constraints of hypernym relations extracted from WordNet to emphasis the hypernym relations into the given word vectors. Whereas Nickel and Kiela [2017] followed a different line of work and proposed Poincaré model that learn hierarchical embeddings into a hyperbolic space, precisely into Poincaré ball, rather than the Euclidean space. Poincaré make use of WordNet hypernymy and merely learn from the taxomomy without a corpus information.

Although the majority of such methods are attractive, because they jointly learn from manually created taxonomies and text corpora, a common disadvantage of such methods is that they mainly focus on pairwise hypernymy relations data in the taxonomies, instead of the entire hierarchical path. A full hierarchical path of hypernymy not only gives a better understanding of the problem than a single hypernymy edge but rather empirically shown to be useful for a pairwise hypernymy identification.

## 3. Hierarchical Word Embeddings (HWE)

We propose a method that learns word embeddings by encoding the hierarchy of hypernymy relations between words. To explain our method, let us consider an example, given a hierarchical hypernym path (*bird → vertebrate → chordate → organism → animal*) where the pairs *(bird,vertebrate)*, *(vertebrate,chordate)*, *(chordate,organism)* and *(organism,animal)* represent a *direct* hypernymy relation, whereas *(bird, chordate)* and *(vertebrate, animal)* form an *indirect* hypernymy. We require our embeddings to be able to encode not only the *direct* hypernymy relations between the hypernym and hyponym words, but also the *indirect* and the full hierarchical hypernym path.

Given a taxonomy $\mathcal{T}$ and a corpus $\mathcal{C}$, we propose a method for learning a hierarchical word embedding $\boldsymbol{w_i} \in \mathbb{R}^d$ for the $i$-th word $w_i \in \mathcal{V}$ in a vocabulary $\mathcal{V}$. We use boldface fonts to denote vectors. We assign two vectors for the word $w_i$, respectively denoting a hyponym $\boldsymbol{w}_i$, or a hypernym $\tilde{\boldsymbol{w}}_i$. The proposed method initialises by making use of a set of hierarchical hypernymy paths, which can be obtained from a taxonomy.

Let us assume that $w_i$ is a leaf node in the taxonomy and $\mathcal{P}(w_i)$ is a path that connects $w_i$ to the root of the taxonomy. If the taxonomy is a tree, then only one such path exists. However, if the taxonomy is a lattice or there are multiple senses of $w_i$ represented by different synsets as in the case of WordNet, we might have multiple paths as $\mathcal{P}(w_i)$. Because a taxonomy by definition arranges concepts in a hierarchical order, we would assume that some of the information contained in a leaf node $w_i$ to be inferred from its parent nodes that fall along the path $\mathcal{P}(w_i)$. Different compositional operators could be used then to infer the semantic representation for $w_i$ using its parents such as a recurrent neural network (RNN). However, for simplicity and computational efficiency, we represent the embedding of a leaf node as the sum of its parents' embeddings. This idea can be formalised into an objective function $J_\mathcal{T}$ for the purpose of learning hierarchical word embedding over the entire vocabulary as follows:

$$J_\mathcal{T} = \frac{1}{2} \sum_{i \in \mathcal{V}} \left\| \boldsymbol{w}_i - \sum_{j \in \mathcal{P}(w_i)} \tilde{\boldsymbol{w}}_j \right\|_2^2 \tag{1}$$

The indirect hypernym at the top of path (i.e. the root of a taxonomy in case it is a tree or the farthest from the hyponym word $w_i$ for a truncated path) represents less (more abstract) information about the word $w_i$ than its direct hypernym. For example, revisiting our hierarchical example above (*bird* $\rightarrow$ *vertebrate* $\rightarrow$ *chordate* $\rightarrow$ *animal*), the direct hypernym *vertebrate* represents more information about $bird$ than the indirect hypernym *animal*. To reflect this in the objective defined in (1), we introduce a discounting term $\lambda(\tilde{w}_j)$ to assign a weight for each hypernym word in the path, and update the objective function given by (1) as follows:

$$J_\mathcal{T} = \frac{1}{2} \sum_{i \in \mathcal{V}} \left\| \boldsymbol{w}_i - \sum_{j \in \mathcal{P}(w_i)} \lambda(\tilde{w}_j) \tilde{\boldsymbol{w}}_j \right\|_2^2 \tag{2}$$

We set $\lambda(\tilde{w}_j) = \exp(\mathcal{L}_{w_i} - \mathcal{D}_{\tilde{w}_j})$ where $\mathcal{L}_{w_i}$ and $\mathcal{D}_{\tilde{w}_j}$ respectively denote the length of the hierarchical hypernymy path of the word $w_i$, and the distance in words between the word $w_i$ and its hypernym $\tilde{w}_j$ in the path, where the distances are measured over the shortest path from the root to word in the taxonomy.

The objective function given by (2) learns the word embeddings purely from the taxonomy $\mathcal{T}$. It does not consider the contextual co-occurrences between the hyponym and hypernym words in the corpus $\mathcal{C}$. To address this problem, for each hypernym word $\tilde{w}_j$ that appears in the path of the hyponym $w_i$, we look up their co-occurrence in the corpus. For this purpose, we first create a co-occurrence matrix $\mathbf{X}$ between the hyponym and hypernym words within a context window in the corpus. The element $X_{ij}$ of $\mathbf{X}$ denotes the total occurrences between the words $w_i$ and $\tilde{w}_j$ in the corpus. We then use the Global Vector (GloVe) [Pennington et al., 2014] objective to consider the co-occurrence between the hyponym word $w_i$ and its hypernyms $\tilde{w}_j$ for the purpose of learning the

embeddings as follows:

$$J_{\mathcal{C}} = \frac{1}{2} \sum_{i \in \mathcal{V}} \sum_{j \in \mathcal{P}(w_i)} f(X_{ij}) \Big( {\boldsymbol{w}_i}^{\top} \tilde{\boldsymbol{w}_j} + b_i + b_j - \log(X_{ij}) \Big)^2 \tag{3}$$

Where $b_i$ and $b_j$ are real-valued scalers biases associated with $w_i$ and $\tilde{w}_j$. The function $f$ is a weighting function defined as follows.

$$f(t) = \begin{cases} (t/t_{\max})^{\alpha} & \text{if } t < t_{\max} \\ 1 & \text{otherwise} \end{cases} \tag{4}$$

Finally, we combine the two objectives given by (2) and (3), into a joint linearly-weighted objective as follows:

$$J = J_{\mathcal{T}} + J_{\mathcal{C}} \tag{5}$$

To minimise (5) w.r.t. the parameters $\boldsymbol{w}_i$, $\tilde{\boldsymbol{w}}_j$, $b_i$ and $b_j$, we compute the gradient of $J$ w.r.t. those parameters. We randomly initialise all parameters and perform Stochastic Gradient Descent with the learning rate scheduled via AdaGrad [Duchi et al., 2011].

## 4. Experiments and Results

We evaluate the embeddings learnt by the proposed **HWE** on three main tasks: the standard supervised hypernymy identification, graded lexical entailment prediction and a newly-proposed hierarchical path completion. In all tasks, we compare the performance of the **HWE** with various prior works on learning word embeddings. In Section 4.2, we first describe the experimental setup that we follow to train the proposed method. We then introduce the word embedding models that we use to compare the proposed method with (Section 4.2). Next, in Section 4.3, we discuss the experiments conducted on the supervised hypernymy identification task. Next, we introduce the experiments of predicting the graded lexical entailment task. We then describe the newly proposed evaluation task, hierarchical path completion (Section 4.5).

### 4.1 Experimental Setup

Any taxonomy can be used as $\mathcal{T}$ with the proposed method provided that the hypernymy relations that exist between words is specified. We use WordNet [Miller, 1995] in our experiments. The average path length of the WordNet words is 7 words. Following the recommendation in prior work on extracting taxonomic relations, we exclude the top-level hypernyms in each path. For example, Anh et al. [2016] found that words such as *object*, *entity* and *whole* at the upper level of the hierarchical path to be noisy during the learning phase. In fact, for the nouns in the WordNet, the word 'entity' is the root for all the words. Moreover, the words like *physical_entity*, *abstraction*, *object* and *whole* appear in the hierarchical path of $58\%$, $47.27\%$, $34.74\%$ and $30.95\%$ of the WordNet words respectively. As such, we limit the number of words in each path to 5 hypernyms, and obtained direct and indirect hypernymy relations. As a results, the number of extracted relations was $256,442$ forming a $59,908$ distinct hierarchical paths, and the vocabulary size $\mathcal{V} = 80,673$.

As a corpus $\mathcal{C}$, we used the ukWaC[1] which has ca. 2 billion tokens. Following the recommendations made in [Levy et al., 2015a], we set the context window to 10 tokens to the both sides of the hyponym word. We followed the GloVe settings and set $\alpha = 0.75$ and $t_{max} = 100$ in (4).

### 4.2 Word Embedding Models

We compare the **HWE** with prior word embedding models that vary from state-of-the-art models that: (i) merely trained on text corpus (Continuous-Bag-Of-Words (**CBOW**), Skip-Gram with Negative Sampling (**SGNS**) [Mikolov et al., 2013] and Global Vector **(GloVe)** [Pennington et al., 2014]) (ii) jointly learn from a taxonomy and a text corpus (**Retrofit**ting [Faruqui et al., 2015], **JointReps** [Bollegala et al., 2016, Alsuhaibani et al., 2018]) and (iii) learn a hypernymy-specific word embeddings (**HyperVec** [Nguyen et al., 2017],**Poincaré** [Nickel and Kiela, 2017],**LEAR** [Vulić and Mrkšić, 2018]).

**CBOW** and **SGNS** are two log-bilinear single-layer neural models proposed by Mikolov et al. [2013] that exploits the *local* co-occurrence between the words in a large text corpus. Where **CBOW** objective is to predict the word given its context words, **SGNS** in contrast predicts the context words given the focus word. **GloVe** is a log-bilinear model that uses the *global* co-occurrences between the word and its context words to predict their embeddings, and it is represented by the objective function given by (3). **Retrofit** and **JointReps** are two models that combine the two resources, taxonomy and text corpus, for learning word embeddings. In **Retrofit**, a pre-trained word embeddings are fed into a taxonomy in a *post-processing* manner, whereas **JointReps** jointly learns the word embeddings from the two resources *simultaneously*. On the other hand, **HyperVec**, **Poincaré** and **LEAR** are hypernymy-specific word embeddings. Details of these models are described in Section 2.

We used the same ukWaC corpus that is used with the proposed method to train the prior methods (**CBOW**, **SGNS**, **GloVe**, **Retrofit**, **JointReps** and **HyperVec**) using the publicly available implementations by the original authors for each method. In all the experiments, we also follow the same settings used with the proposed method, and set the context window to 10 words to both sides of the focus word, and remove the words that appear less than 20 times in the corpus. Following Levy et al. [2015a], we set the negative sampling rate to 5 in **SGNS**. We retrofit the embeddings learnt by **SGNS** and **CBOW** into the **Retrofit** model. Moreover, for **Retrofit**, **JointReps** and **HyperVec**, we used the *hypernym* relations extracted from the WordNet provided by the original authors for each model to derive the taxonomy constraints during the training. For **Poincaré** model, we used the gensim implementation [Řehůřek and Sojka, 2010] and set the negative samples to 10. In **LEAR**, we used the pre-trained embeddings provided by the original authors. We learn a 300 dimensional word embeddings in all experiments.

### 4.3 Supervised Hypernymy Identification

The supervised hypernymy identification is a standard task to evaluate the ability of the word embeddings to detect hypernymy. The task is modeled as a binary classification problem, where a classifier is trained using pairs of words $(x, y)$ labeled as *positive* (i.e. a hypernymy relation exists between the words) or *negative* (otherwise). Each word in a word-pair is represented by its pre-trained word embedding. Several approaches have been proposed to combine the embeddings of the two words in each word-pair to create a single feature vector representing the relation between

---

1. http://wacky.sslmit.unibo.it

the two words that is subsequently given to the classifier. For example, *concatenation* $(\boldsymbol{x} \oplus \boldsymbol{y})$ [Baroni et al., 2012], *difference* $(\boldsymbol{x} - \boldsymbol{y})$ and *addition* $(\boldsymbol{x} + \boldsymbol{y})$ [Weeds et al., 2014] have been used as operators for creating feature vectors for word-pairs. In our preliminary experiments, we found that *concatenation* to perform best among all of those operators for the task of training a hypernymy identification. Therefore, we use *concatenation* as the preferred operator in the remainder of the experiments described in the paper. We use a binary Support Vector Machine (SVM) classifier, implemented in scikit-learn [Pedregosa et al., 2011]. We set an RBF kernel $\gamma = 0.03125$ and a cost parameter $C = 8.0$, using an independent validation dataset.

| Dataset | #Instances | Ratio pos/neg |
|---------|------------|---------------|
| Kotlerman | 2,940 | 0.42 |
| Bless | 14,547 | 0.11 |
| Baroni | 2,770 | 0.98 |
| Levy | 12,602 | 0.08 |

Table 1: Summary of the datasets used in the supervised hypernymy identification task.

We pick four widely used hypernym benchmark datasets, **Kotlerman** [Kotlerman et al., 2010], **Bless** [Baroni and Lenci, 2011], **Baroni** [Baroni et al., 2012] and **Levy** [Levy et al., 2014], for the supervised classification task. To avoid any *lexical memorisation*, where the classifier simply *memorises* the prototypical hypernyms rather than *learning* the relation, [Levy et al., 2015b] introduced a disjoint version with none lexical overlap between the test and train portions for each of the above datasets, we adopt these disjoint versions in our experiment. Details of each benchmark dataset is summarised in Table 1.

In Table 2, we compare the quality of our embeddings (**HWE**) against that of the other methods on the hypernym identification benchmarks. Although F1 score has been widely-used as an evaluation measure in prior work on hypernym identification, [Sanchez and Riedel, 2017] argued that Area Under the ROC Curve (**AUC**) is more appropriate as an evaluation measure for this task because some of the benchmark datasets are unbalanced in terms of the number of positive vs. negative test instances they contain. To be able to compare our results with prior work that report only F1 score, we use both F1 score as well as AUC as evaluation measures in our experiments.

From Table 2, we see that the **HWE** reports the best scores in two of the benchmark datasets. In particular, in **Levy** dataset, **HWE** reports the best performance, with a slight improvement over the first two categories of methods in the table, but significant improvement (binomial test, $p < 0.05$) over the closest related works in the third category. Similarly, **HWE** scores the highest in the **Baroni** dataset where we can observe a strong difference between the hypernymy-specific word embedding models and other methods. In particular, **HyperVec**, **LEAR** and **HWE** significantly outperforms other methods and **HWE** reports the best score in this dataset. This result is particularly noteworthy, because a prior extensive analysis on different benchmark datasets for hypernym identification by Sanchez and Riedel [2017] concluded that the **Baroni** dataset is the most appropriate dataset for robustly evaluating hypernym identification methods. In **Kotlerman** and **Bless** datasets, **LEAR** reports the best scores among all models. However, Table 2 shows that even the methods that were trained only with a text corpus without specifically designed to capture the hypernymy relation, perform well in the **Bless** dataset, reporting a better or a comparable performance to the hypernymy-specific embeddings such as **HWE** and **HyperVec**. It is worth noting that both of the closest related

| Model | Kotlerman | | Bless | | Baroni | | Levy | |
|---|---|---|---|---|---|---|---|---|
| | F1 | AUC | F1 | AUC | F1 | AUC | F1 | AUC |
| CBOW | 56.07 | 56.20 | 89.58 | 86.89 | 68.02 | 68.41 | 68.58 | 64.23 |
| SGNS | 56.17 | 56.06 | 88.80 | 86.46 | 68.25 | 68.61 | 68.34 | 64.53 |
| GloVe | 56.26 | 56.49 | 89.92 | 88.59 | 69.86 | 70.27 | 68.07 | 64.61 |
| Retrofit(CBOW) | 53.82 | 56.40 | 84.78 | 78.93 | 70.34 | 70.69 | 50.28 | 51.11 |
| Retrofit(SGNS) | 53.27 | 56.11 | 84.25 | 78.32 | 71.52 | 71.84 | 49.48 | 50.73 |
| JointReps | 56.33 | 56.40 | 90.76 | 89.64 | 69.74 | 70.16 | 67.76 | 64.66 |
| HyperVec | 57.16 | 57.33 | 87.78 | 83.86 | 76.80 | 76.61 | 54.71 | 54.94 |
| Poincaré | 45.66 | 51.56 | 47.13 | 50.00 | 54.96 | 55.56 | 48.02 | 50.00 |
| LEAR | **64.62** | **64.14** | **93.46** | **91.83** | 77.26 | 77.68 | 66.82 | 62.84 |
| HWE | 57.32 | 57.63 | 86.06 | 81.97 | **77.37** | **77.73** | **68.59** | **64.83** |

Table 2: Classifier performance using different embedding methods as features on several hypernym benchmark datasets.

works, **HyperVec** and **LEAR**, use pairwise hypernym relations in a similar spirit to the structure of the benchmark datasets, whereas the **HWE** use the entire hierarchical path, nevertheless reporting a better or comparable results.

## 4.4 Graded Lexical Entailment

An important aspects of the **HWE** embeddings is its ability to encode the hierarchical structure available in the taxonomy in the learnt embeddings. To further check this ability we use the gold standard dataset **HyperLex** [Vulić et al., 2017] to test how well the **HWE** embeddings capture graded lexical entailment. HyperLex focuses on the relation of graded or soft lexical entailment at a continuous scale rather than simplifying the judgements into a binary decision. The HyperLex dataset consists of 2616 word pairs where each pair is manually annotated with a score on a scale of $[0, 10]$ indicating the strength of the relations of lexical entailment. Lexical entailment is asymmetric in general, therefore we need an asymmetric distance function that takes into account both vector norm and direction to provide correct entailment scores between word pairs. For this purpose, given a hypernymy pair $(x, y)$, where $x$ is the hyponym word and $y$ is the hypernym, we follow Vulić and Mrkšić [2018] and compute the entailment score as follows:

$$\text{score}(\boldsymbol{x}, \boldsymbol{y}) = \text{dcos}(\boldsymbol{x}, \boldsymbol{y}) + (|\boldsymbol{x}| - |\boldsymbol{y}|) \tag{6}$$

where $\text{dcos}(\boldsymbol{x}, \boldsymbol{y})$ is the cosine distance[2]. Following the standard approach for evaluating using the **HyperLex** dataset, we measure Spearman $\rho$ correlation coefficient between gold standard ratings and the predicted scores. Table 3 shows the results of the Spearman correlation coefficients of **HWE** and the other word embeddings models on the **HyperLex** dataset against the human ratings. Table 3 shows that **HWE** was able to encode the hierarchical structure in the learnt embeddings reporting a better or comparable results to all other models except for **LEAR**. In particular, **HWE** reports

---

2. $\text{dcos}(\boldsymbol{x}, \boldsymbol{y}) = 1 - \cos(\boldsymbol{x}, \boldsymbol{y})$

an improvement over the corpus-based and joint methods as well as **Poincaré** embeddings. It is noteworthy that **LEAR** was trained with larger resources which allows the model to benefit from a large-scale vocabulary, as we use the provided pre-trained embeddings of **LEAR**. We aim to re-run the experiment in the future and use the same resources, text corpus and taxonomy data, that were used with all the models with **LEAR**.

| Model | Spearman's $\rho$ |
|---|---|
| CBOW | 0.11 |
| SGNS | 0.11 |
| GloVe | 0.23 |
| Retrofit(CBOW) | 0.07 |
| Retrofit(SGNS) | 0.13 |
| JointReps | 0.08 |
| HyperVec | 0.47 |
| Poincaré | 0.35 |
| LEAR | **0.63** |
| HWE | 0.46 |

Table 3: Results of HWE and other word embeddings models on the HyperLex dataset.

## 4.5 Hierarchical Path Completion

Considering that our HWEs are designed to capture hierarchical information about words, it remains an open question as to what extent the word embeddings learnt using the proposed method encode the hierarchical structure. The supervised hypernymy identification and the graded lexical entailment tasks discussed in the previous sections (4.3 and 4.4) provides only a partial answer to this question because all the benchmarks used in the experiments there were specifically annotated for hypernymy relation between two words, ignoring the taxonomic structure.

Unfortunately, none of the prior work has looked into this aspect of word embeddings and to the best of our knowledge there exists no benchmark dataset for conducting such a hierarchical evaluation over a taxonomy. To address this issue, we create a novel dataset by first sampling paths from the WordNet, which connect a hypernym to a hyponym via a path not exceeding a maximum path length $\mathcal{L}_{\max}$. We limit the paths to contain words that are unigrams, bigrams or trigrams, and sample the paths such that words with different frequency ranges are covered, thereby considering paths that contain frequent as well as rare words. This enables us to conduct a fair and a balanced evaluation of the hierarchical word embeddings. We remove any paths that are used as training data when computing $\mathcal{J}_c$ in (1). By following this procedure we created a dataset that contains 330 paths. If a path contains only unigrams then it is considered as a unigram path. If a path contains at least one bigram but no trigrams, we consider it to be a bigram path and if a path contains a trigram then it is considered as a trigram path. There are respectively 150, 120 and 60 unigram, bigram and trigram paths in the created dataset.

Inspired by the word analogy prediction task that is widely used to evaluate word embeddings [Mikolov et al., 2013], we propose a *hierarchical path completion* task as follows. Given a hierarchical path $(a \rightarrow b \rightarrow c \rightarrow d \rightarrow e)$ where $a$ is the hyponym word and $b$, $c$, $d$ and $e$ are its

hypernyms, the task is to predict the hyponym word $a$ given its hypernyms $b$, $c$, $d$ and $e$. However, in the taxonomy, we might observe several hyponym words that have the exact same hierarchical path. For example, the hypernyms $(b \rightarrow c \rightarrow d \rightarrow e)$ might represent the hierarchical path to both $a$ and $a'$. In fact, for each hierarchical path in the WordNet, there are, on average, 8 hyponym words represented by that particular hierarchical path. Therefore, for each hierarchical path instance, if a method predicts a candidate $a$ that belongs to any of the hyponyms with that path, we consider it to be a correct prediction.

Given a set of word embeddings learnt using some word embedding learning method, to solve a hierarchical path completion problem, we used several prediction methods to predict a hyponym word $a$ using the hypernyms $b$, $c$, $d$ and $e$ as described next.

**ADD** : We used an *additive* method (**ADD**), in which we compute the cosine similarity between $b + c + d + e$ and each word $x$ in a fixed vocabulary, and select the word $a^*$ with the highest cosine similarity as the prediction. If $a^*$ equals to $a$ (or $a$'s), then we consider it to be a correct prediction for the given test instance.

**SUB** : In addition to the **ADD** method, and following the same procedure of computing the cosine similarity, we further consider several other prediction methods such as $b + c + d - e$, $b + c - d + e$, $b - c - d - e$ and $b - c + d + e$, for predicting $a$ using the hypernyms $b$, $c$,$d$ and $e$. We refer to these methods as the *subtraction* (**SUB**). We found that the variant $b + c + d - e$ of **SUB** to perform consistently better than other **SUB** variations with all embedding models. Considering that the further the subtracted candidate in the hierarchical path from $a$ the lesser its influence on the prediction. For example, the variant $b + c + d - e$ reports a better prediction result as compared to $b - c + d + e$, presumably because of the subtracted hypernym $c$ being closer to the hyponym $a$ as compared to the farthest hyponym $e$. The remainder of the experiments in the paper use the variant $b + c + d - e$ as **SUB**.

**DH** : In this method, we use only the direct hypernym $b$ to predict $a$ (or $a$'s), and refer to this method as the *direct hypernym* (**DH**). We follow the same procedure of computing the cosine similarity as used in the **ADD** and **SUB** methods.

If the predicted candidate is in the set of hyponyms for a particular test hierarchical path, then we consider it to be a correct prediction. We then define the *accuracy* of a prediction method that uses word embeddings learnt from a particular word embedding learning method as the percentage of the correctly answered test instances.

In Table 4, we report the accuracies of different word embedding learning methods on the hierarchical path completion dataset using different prediction methods over different $n$-gram lengths. According to Clopper-Pearson confidence intervals computed at $p < 0.05$, the proposed **HWE** significantly outperforms all other word embedding learning methods compared in Table 4 irrespective of the prediction method being used.

Surprisingly, in contrast to the results found on the supervised classification task (4.3) and the graded lexical entailment task (4.4), where the prior word embeddings methods, including hypernymy-specific methods **HyperVec** and **LEAR** were performing constantly well on pairwise datasets, they seem to suffer capturing an upper level hierarchy in the vector space. Whereas, on other side, **Poincaré** which was suffering with pairwise data in the previous tasks, seems to learn a better upper level hierarchy and works well with this hierarchical completion task. The fact that **Poincaré** embeddings, a hierarchical word embeddings learn method, reports good performance

| Model | Prediction Method | | | | | | | | |
|---|---|---|---|---|---|---|---|---|---|
| | ADD | | | SUB | | | DH | | |
| | Uni | Uni+Bi | Uni+Bi+Tri | Uni | Uni+Bi | Uni+Bi+Tri | Uni | Uni+Bi | Uni+Bi+Tri |
| CBOW | 38.25 | 36.96 | 35.18 | 36.43 | 35.74 | 35.22 | 44.43 | 44.38 | 43.16 |
| SGNS | 32.77 | 31.24 | 29.70 | 33.68 | 33.05 | 32.59 | 40.95 | 40.27 | 39.24 |
| GloVe | 29.60 | 27.59 | 25.66 | 37.56 | 27.42 | 26.90 | 44.18 | 42.83 | 42.66 |
| Retrofit(CBOW) | 38.58 | 38.32 | 37.82 | 37.67 | 37.28 | 36.76 | 44.94 | 43.39 | 43.23 |
| Retrofit(SGNS) | 34.73 | 33.95 | 33.27 | 34.73 | 34.09 | 33.85 | 42.92 | 42.14 | 41.88 |
| JointReps | 31.19 | 30.85 | 29.65 | 35.69 | 34.51 | 34.16 | 40.02 | 39.68 | 39.31 |
| HyperVec | 19.09 | 18.58 | 18.24 | 25.45 | 25.16 | 24.72 | 36.36 | 35.80 | 35.17 |
| Poincaré | 60.91 | 58.24 | 58.17 | 59.09 | 57.70 | 57.67 | 60.92 | 59.08 | 58.66 |
| LEAR | 25.45 | 25.22 | 24.73 | 22.91 | 20.25 | 20.06 | 28.45 | 27.77 | 26.60 |
| HWE | **88.30** | **85.19** | **84.75** | **71.39** | **69.89** | **69.65** | **84.96** | **81.28** | **81.05** |

Table 4: Accuracy (%) of the different word embedding learning models on the hierarchical path completion dataset using the ADD, SUB and DH as prediction methods over **Uni**gram, **Bi**gram and **Tri**gram paths.

on this hierarchical path completion task, suggests that this is an appropriate task for evaluating hierarchies and embeddings.

Using the **ADD** method, **HWE** reports an improvement as high as $28\%$ on accuracy over **Poincaré**, which is the highest score among the other methods. Table 4 shows that **SUB** slightly increases the other word embedding methods' results and decreases the performance of **HWE**. **DH** significantly improves the results for all word embeddings, except for **HWE** and **Poincaré**, whereas **ADD** applied to **HWE** reports the best overall performance in Table 4. This result shows that the proposed **HWE** not relying merely on the direct hypernym but considers information available in the entire hierarchical path composed into a single vector via addition. Furthermore, Table 4 shows that including *bigram* and *trigram* hypernyms on the paths slightly hurt the performance, possibly because of data sparseness.

To further demonstrate the ability of the proposed method for completing the hierarchical paths, we qualitatively analyse the predictions of **HWE** and **Retrofit(CBOW)**. Due to the limited availability of space, only a few randomly selected examples are shown in Table 5. The hyponym column represents gold standard answers (i.e. correct hyponym words). Due to space limitations, we show only a maximum of $4$ correct hyponyms in the table. If a particular path has more than $4$ hyponyms, we randomly select $4$, otherwise all possible hyponyms are listed. We see that **HWE** accurately predicts the correct word in many cases where **Retrofit(CBOW)** fails (shaded rows in the table). We can also see **Retrofit(CBOW)** tends to predict the average nearest neighbour of a word's hypernyms to be the missing word, not necessarily completing the hierarchical path. For example, given the path (*? → head_dress → clothing → consumer_goods → commodity*), **HWE** correctly predicts the missing word to be *hat*, whereas **Retro(CBOW)** incorrectly predicts *dress*. More interestingly, Table 5 shows by considering the hierarchical path we can reduce the ambiguity of the prediction. For instance, given the path (*? → book → publication → work → product*) the **Retrofit(CBOW)** predicts

| Hypernym$_1$ (b) | Hypernym$_2$ (c) | Hypernym$_3$ (d) | Hypernym$_4$ (e) | Hyponym(s) (a's) | HWE Prediction | Retrofit Prediction |
|---|---|---|---|---|---|---|
| book | publication | work | product | booklet,textbook, storybook,catalog | textbook | nature |
| success | attainment | accomplishment | action | winning, flying_colors | winning | enjoyment |
| head_dress | clothing | consumer_goods | commodity | turban,cap, kaffiyeh,hat | hat | dress |
| supplier | businessperson | capitalist | person | recruiter,distributor, stockist,caterer | distributor | market |
| great_ape | anthropoid_ape | ape | primate | orangutan, gorilla,chimpanzee | gorilla | chimp |
| executive | administrator | head | leader | corporate_executive minister,rainmaker | minister | chair |
| forced_landing | aircraft_landing | landing | arrival | crash_landing | crash_landing | landing |
| placental | mammal | vertebrate | chordate | ungulate,carnivore, lagomorph,rodnet | carnivore | ecology |
| despair | feeling | state | attribute | discouragement, pessimism | discouragemen | apathy |
| waste | activity | act | event | waste_of_effort, boondoggle | boondoggle | enjoyment |
| bank_withdraw | withdrawal | removal | separation | bank_run | bank_run | withdrawal |
| capture | felony | crime | transgression | abduction, kidnapping | kidnapping | kidnapping |
| text | matter | writing | communication | stanza,lyric, lipogram | card | lyric |
| ocean_trip | water_travel | travel | motion | voyage,cruise maiden_voyage | round_trip | voyage |

Table 5: Selected predictions of HWE and Retrofit(CBOW) on the hierarchical path completion task (ADD). Hyponym(s) represents gold standard answer(s).

the word *nature* to complete the path, presumably associating this with the scientific journal *Nature*[3] because the path includes *publication*, however **HWE** correctly predicted the word to be *textbook*. Moreover, from Table 5, we can see that in some cases **HWE** struggled to predict the correct words, while **Retrofit(CBOW)** has managed to accurately complete the path. For example, **HWE** failed to predict the word(s) *voyage,cruise* or *maiden_voyage* to complete the path ($? \rightarrow ocean\_trip \rightarrow world\_travel \rightarrow travel \rightarrow motion$) while **Retrofit(CBOW)** was able to do so. Considering the fact that *ghost* and *apparition* are registered as synonyms in the WordNet, **Retrofit(CBOW)** could be simply picking up this synonym, without considering any hierarchical information in this case.

Furthermore, we study the effect of each hypernym word in the hierarchical path towards the prediction accuracy. In particular, given a hierarchical path ($a \rightarrow b \rightarrow c \rightarrow d \rightarrow e$), the goal is to study which of the given hypernyms $b$, $c$ , $d$ and $e$ is having the strongest effect on predicting the correct word $a$. For this purpose, we used the same hierarchical path completion dataset including *bigram* and *trigram* hypernyms and follow the same procedure used in the above experiment, but we hold out one of the hypernym while computing the cosine similarity between $b+c+d+e$ and $a$. We vary the excluded hypernym between $b$, $c$, $d$ and $e$. When we hold out $b$ we refer to this as the direct

---

3. www.nature.com

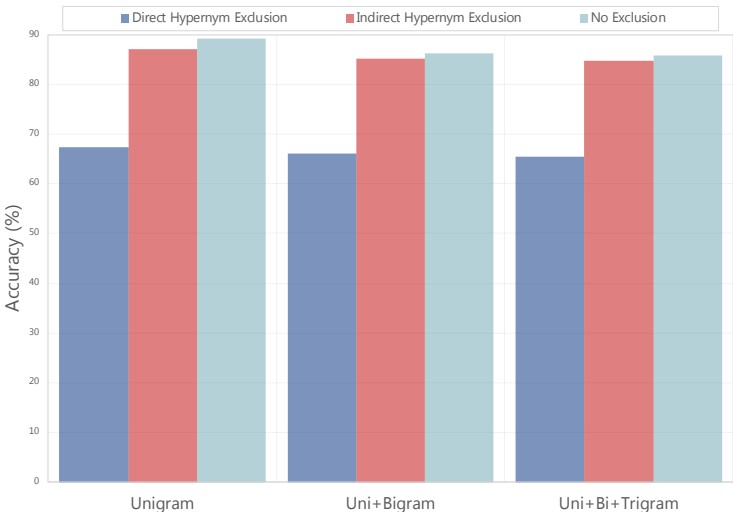

Figure 1: Comparison of direct and indirect hypernym exclusion from a word's path evaluated on the hierarchical path completion dataset.

hypernym exclusion, whereas holding out one of $c$, $d$ or $e$ is referred to as the indirect hypernym exclusion. In Figure 1, we report the accuracy result on the hierarchical path completion task using the **ADD** method with each exclusion variation. Not surprisingly, excluding the direct hypernym ($b$) drastically effect the performance. Whereas, the least effect can be observed with the farthest hypernym ($e$). For example, in a hierarchical path of a hyponym word of only *uigram* hypernyms, excluding the first direct hypernym results on $68\%$ accuracy prediction, whereas holding out the farthest hypernym significantly improves the prediction score reaching $88\%$ accuracy.

We also investigate how the dimensionality $d$ effects the proposed method. Similar to the previous experiments, we report the accuracy of predicting the hyponym word in each hierarchical hypernym path. From Figure 2, we see that the proposed method is able to reach as high as $76\%$ with as small as 25 dimensions. The performance is then increase with the dimensionality, reaching its peak with 200 dimensions reporting $88\%$ accuracy. It is worth noting that, adding more dimensions does not negatively effect the performance. Moreover, Figure 2 shows that including *bigram* and *trigram* hypernym words in the paths report a slight decrease in the performance, but similar trend as to the *unigram only* is observed.

## 5. Conclusion

We presented a method to learn a Hierarchical Word Embedding (**HWE**) for identifying the hypernymy relations between words. For this purpose, we introduced a joint objective that make use of both a taxonomy and a large text corpus to learn hierarchical word embeddings. We evaluated **HWE** on the standard supervised hypernymy identification and a newly-proposed hierarchical hypernymy path completion tasks. The experiments conducted in this paper on the above mentioned two tasks demonstrate that **HWE** was able to encode the hypernymy relations between words into the learnt embeddings, and reports an improvement over several previously proposed methods that learn either general word embeddings or hypernymy-specific word embeddings.

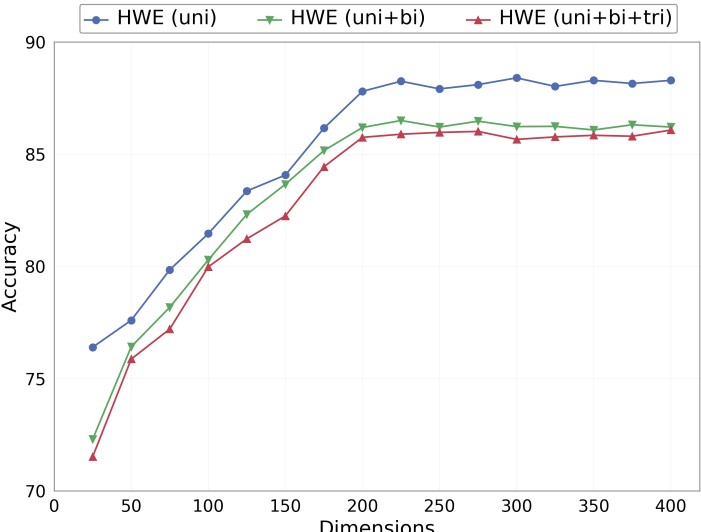

Figure 2: Effect of dimensions on the proposed method evaluated on hierarchical path completion dataset.

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
