# OpenReview forum: "Joint Learning of Hierarchical Word Embeddings from a Corpus and a Taxonomy"
_AKBC.ws/2019/Conference — AKBC 2019_

### Official Review · AnonReviewer2 · 2019-01-09
**Needs better evaluation**

**Rating:** 5
**Confidence:** 4

**Review:**

This paper presents a method to jointly learn word embeddings using co-occurrence statistics as well as by incorporating hierarchical information from semantic networks like WordNet.

In terms of novelty, this work only provides a simple extension to earlier papers [1,2] by changing the objective function to instead make the word embeddings of a hypernym pair similar but with a scaling factor that depends on the distance of the words in the hierarchy.

While the method seems to learn some amount of semantic properties, most of the baselines reported seem either outdated or ill fitted to the task and do not serve well to evaluate the value of the proposed method for the given task.
For example the JointRep baseline is based on a semantic similarity task which primarily learns word embeddings based on synonym relations and seems to not be an appropriate baseline to compare the current approach to.
Further, there are two primary methods of incorporating semantic knowledge into word embeddings - by incorporating them during the training procedure or by post processing the vectors to include this knowledge. While I understand that this method falls into the first category, it is still important and essential to compare to both types of strategies of word vector specialization. In this regard [3]  has been shown to beat HyperVec and other methods on hypernym detection and directionality benchmarks and should be included in the results. It would be also interesting to see how the current approach fares on graded hypernym benchmarks such as Hyperlex.

Minor comments : Section 4.2 there is a word extending out of the column boundaries.


[1] Alsuhaibani, Mohammed, et al. "Jointly learning word embeddings using a corpus and a knowledge base." PloS one (2018)
[2] Bollegala, Danushka, et al. "Joint Word Representation Learning Using a Corpus and a Semantic Lexicon." AAAI. 2016.
[3] Vulić, Ivan, and Nikola Mrkšić. "Specialising Word Vectors for Lexical Entailment." NAACL-HLT 2018.

---

> ### Author Response · Authors · 2019-01-31
> **More evaluation task has been added.**
>
>
> -->Q1: While the method seems to learn some amount of semantic properties, most of the baselines reported seem either outdated or ill fitted to the task and do not serve well to evaluate the value of the proposed method for the given task.
>
> Ans: We have now updated the paper with two more recent relative work, Poincare[1] and LEAR[2], and empirically compare the proposed method against them in all the three evaluation tasks.
>
> [1] Poincare Embeddings [Nikel and Kiela] - NIPS 2017
> [2] Specialising Word Vectors for Lexical Entailment [Vulic and Mrksic] - NAACL-HLT 2018
>
> -->Q2: For example the JointRep baseline is based on a semantic similarity task which primarily learns word embeddings based on synonym relations and seems to not be an appropriate baseline to compare the current approach to.
>
> Ans: The JointReps method used different semantic relations (synonyms, hypernyms, hyponyms… etc), however, here we used the hypernym relation when we train the JointReps to compare the proposed method with it in all the evaluation tasks.
>
> -->Q3: Further, there are two primary methods of incorporating semantic knowledge into word embeddings - by incorporating them during the training procedure or by post processing the vectors to include this knowledge. While I understand that this method falls into the first category, it is still important and essential to compare to both types of strategies of word vector specialization.
>
> Ans: The evaluation proposed in the paper does, in fact, compare against both type of categories. JointReps and HyperVec fall into the first category, whereas the Retrofit method and the newly added (in the updated version of the paper) LEAR method falls into the second. In all of these methods, we used the hypernym relations to incorporate the semantic knowledge into the learnt embeddings.
>
> -->Q4: In this regard [3] has been shown to beat HyperVec and other methods on hypernym detection and directionality benchmarks and should be included in the results.
>
> Ans: Thank you for the suggestion. We have now added LEAR to the updated version of the paper and empirically compare the proposed method against in all the three evaluation tasks.
> The proposed method reports better or comparable results to LEAR in two of the main evaluation tasks (hypernym detection and hierarchical completion).
>
> -->Q5: It would be also interesting to see how the current approach fares on graded hypernym benchmarks such as Hyperlex.
>
> Ans: Thank you for the suggestion. We have added a new sub-section (section 4.4) in the updated version of the paper with a new evaluation task on the graded lexical entailment prediction using HyperLex.
>
> -->Q6: Section 4.2 there is a word extending out of the column boundaries.
>
> Ans: Thank you. This has been modified in the updated version of the paper.

---

### Official Review · AnonReviewer1 · 2019-01-09
**Well motivated approach but some concerns**

**Rating:** 5
**Confidence:** 4

**Review:**

This paper proposed a joint learning method of hypernym from both raw text and supervised taxonomy data.  The innovation is that the model is not only modeling hypernym pairs but also the whole taxonomy. The experiments demonstrate better or similar performance on hypernym pair detection task and much better performance on a new task called "hierarchical path completion". The method is good motivated and intuitive. Lots of analysis on the results are done which I liked a lot. But I have some questions for the authors.

1) One major question I have is for the taxonomy evaluation part, I think there are works trying to do taxonomy evaluation by using node-level and edge-level evaluation. 'A Short Survey on Taxonomy Learning from Text Corpora:
Issues, Resources, and Recent Advances' from NAACL 2017 did a nice summarization for this. Is there any reason why this evaluation is not applicable here?

2) At the end of section 4.2, the author mentioned Retrofit, JointReps and HyperVec are using the original author prepared wordnet data. Then the supervised training data is different for different methods? Is there a more controlled experiment where all experiments are using the same training data?

3) In section 4.4, there are three prediction methods are introduced including ADD, SUM, and DH. The score is calculated using cosine similarity. But the loss function used in the model is by minimizing the L2 distance between word embeddings? Is there any reason why not use L2 but cosine similarity in this setting? Also, I'm assuming SUM and DH are using cosine similarity as well? It might be useful to add that bit of information.

4) The motivation for this paper is to using taxonomy instead of just hypernym pairs? Another line of research trying to encode the taxonomy structure into the geometry space such that the taxonomy will be automatically captured due to the self-organized geometry space. Some papers including but not restricted 'Order-Embeddings of Images and Language',
'Probabilistic Embedding of Knowledge Graphs with Box Lattice Measures'. Probably this line of work is not directly comparable, but it might be useful to add to the related work session.

A few minor points:
1) In equation four of section 3, t_max appears for the first time. This equation maybe part of the GLOVE objective, but a one-sentence explanation of t_max might be needed here.
2) at the end of section 3, the calculation of gradients for different parameters are given, but the optimization is actually performed by AdaGrad. Maybe it would be good to move these equations to the appendix.
3) In section 4.1 experiment set up, the wordnet training data is generated by performing transitive closure I assume? How does the wordnet synsets get mapped to its surface form in order to do further training and evaluation?

---

> ### Author Response · Authors · 2019-01-31
> **Concerns clarification**
>
> -->Q1: One major question I have is for the taxonomy evaluation part, I think there are works trying to do taxonomy evaluation by using node-level and edge-level evaluation. 'A Short Survey on Taxonomy Learning from Text Corpora:
> Issues, Resources, and Recent Advances' from NAACL 2017 did a nice summarization for this. Is there any reason why this evaluation is not applicable here?
>
> Ans: The above-mentioned paper is mainly about the recent work on taxonomy construction from free texts, which is different from what we are proposing in this paper. Our goal is not to create taxonomies but to learn word embeddings that preserve taxonomic information as vector representations. As we do not create taxonomies, we cannot evaluate the word embeddings using taxonomy evaluation methods.
>
> -->Q2: At the end of section 4.2, the author mentioned Retrofit, JointReps and HyperVec are using the original author prepared wordnet data. Then the supervised training data is different for different methods? Is there a more controlled experiment where all experiments are using the same training data?
>
> Ans: The models Retrofit, JointReps, and HyperVec (and the newly added two recent relevant work (Poincare and LEAR)) work with pairwise relation data. However, the proposed HWE works on a full hierarchical hypernym path. Therefore, the models require slightly different data.
>
> -->Q3: In section 4.4, there are three prediction methods are introduced including ADD, SUM, and DH. The score is calculated using cosine similarity. But the loss function used in the model is by minimizing the L2 distance between word embeddings? Is there any reason why not use L2 but cosine similarity in this setting?
>
> Ans: We have empirically tested both the L2 and cosine and found that the cosine to work better in the given experiment.
>
> -->Q4: Also, I'm assuming SUM and DH are using cosine similarity as well? It might be useful to add that bit of information.
>
> Ans: Yes. This has been added in the updated version of the paper.
>
> -->Q5: The motivation for this paper is to using taxonomy instead of just hypernym pairs?
>
> Ans: Yes
>
> -->Q6: Another line of research trying to encode the taxonomy structure into the geometry space such that the taxonomy will be automatically captured due to the self-organized geometry space. Some papers including but not restricted 'Order-Embeddings of Images and Language', 'Probabilistic Embedding of Knowledge Graphs with Box Lattice Measures'. Probably this line of work is not directly comparable, but it might be useful to add to the related work session.
>
> Ans: Thank you for the suggestion. We have now updated the paper with two more related works, Poincare[1] and LEAR[2] and empirically compare the proposed method against them.
> Please note that Probabilistic and Box Embeddings are relatively less related to the proposed method as they working on phrase embeddings and used pre-trained word embeddings to feed an LSTM for learning phrase embeddings.
>
> [1] Poincare Embeddings [Nikel and Kiela] - NIPS 2017
> [2] Specialising Word Vectors for Lexical Entailment [Vulic and Mrksic] - NAACL-HLT 2018
>
> -->Q7: In equation four of section 3, t_max appears for the first time. This equation maybe part of the GLOVE objective, but a one-sentence explanation of t_max might be needed here.
>
> Ans: Yes, it is the weighting function of GloVe so that it becomes relatively small for words of large frequency and set to 100  as stated in section (4.1).
>
> -->Q8: at the end of section 3, the calculation of gradients for different parameters are given, but the optimization is actually performed by AdaGrad. Maybe it would be good to move these equations to the appendix.
>
> Ans: The gradients equations have been moved to the appendix in the updated version of the paper.
>
> -->Q9: In section 4.1 experiment set up, the wordnet training data is generated by performing transitive closure I assume? How does the wordnet synsets get mapped to its surface form in order to do further training and evaluation?
>
> Ans: We lemmatise the corpus and use the form given in the WordNet as the surface form.

---

### Official Review · AnonReviewer3 · 2019-01-10
**Recent relevant work not adequately discussed or compared.**

**Rating:** 4
**Confidence:** 4

**Review:**

Paper summary:  This paper presents a method of learning word embeddings for the purpose of representing hypernym relations.  The learning objective is the sum of (a) a measure of the “distributional inclusion” difference vector magnitude and (b) the GloVE objective.  Experiments on four benchmark datasets are mostly (but not entirely positive) versus some other methods.

The introduction emphasizes the need for a representation that "able to encode not only the direct hypernymy relations between the hypernym and hyponym words, but also the indirect and the full hierarchical hypernym path.”  There has been significant interest in recent work on representations aiming for exactly this goal, including Poincare Embeddings [Nikel and Kiela], Order Embeddings [Vendrov et al], Probabilistic Order Embeddings [Lai and Hockenmaier], Box embeddings [Vilnis et al].  It seems that there should be empirical comparisons to these methods.

I found the order of presentation awkward, and sometimes hard to follow.  For example, I would have liked to see a clear explanation of test-time inference before the learning objective was presented, and I’m still left wondering why there is not a closer correspondence between the multiple inference methods described (in Table 3) and the learning objective.

I would also have liked to see a clear motivation for why the GloVE embedding is compatible with and beneficial for the hypernym task.  “Relatedness” is different than “hypernymy.”

---

> ### Author Response · Authors · 2019-01-31
> **More recent relevant work have been added**
>
> -->Q1: There has been significant interest in recent work on representations aiming for exactly this goal, including:
> Poincare Embeddings [Nikel and Kiela],
> Order Embeddings [Vendrov et al],
> Probabilistic Order Embeddings [Lai and Hockenmaier],
> Box embeddings [Vilnis et al].
> It seems that there should be empirical comparisons to these methods.
>
> Ans: Thank you for the suggestion. Poincare seems to be an excellent fit to be empirically compared against the proposed method, as they both share a similar spirit to explicitly learn hierarchical word embeddings rather than hypernymy-specific embeddings
> We have now updated the paper with two more related works, Poincare[1] and LEAR[2] and empirically compare the proposed method against them.
> We have also updated the paper with a new evaluation task (section 4.4) to test the proposed method on a graded lexical entailment as suggested by reviewer3.
> The proposed method reports an improvement over most of the prior works, including Poincare, in the three tasks (hypernym detection, graded lexical entailment, and the hierarchical path completion), except for the LEAR in two datasets.
> More interestingly, Poincare seems to perform well in the proposed hierarchical path completion task in contrast to the other methods apart from HWE. The fact that Poincare embeddings, a hierarchical word embeddings learn method, reports good performance on this hierarchical path completion task, suggests that this is an appropriate task for evaluating hierarchies and embeddings.
>
> Please note that Probabilistic and Box Embeddings are relatively less related to the proposed method as they work on phrase embeddings and use pre-trained word embeddings to feed an LSTM for learning phrase embeddings.
>
> [1] Poincare Embeddings [Nikel and Kiela] - NIPS 2017
> [2] Specialising Word Vectors for Lexical Entailment [Vulic and Mrksic] - NAACL-HLT 2018
>
>
> -->Q2: I found the order of presentation awkward, and sometimes hard to follow. For example, I would have liked to see a clear explanation of test-time inference before the learning objective was presented, and I’m still left wondering why there is not a closer correspondence between the multiple inference methods described (in Table 3) and the learning objective.
>
> Ans: We use the hierarchical word embeddings produced by the proposed method (HWE) in three tasks: hypernym detection (section 4.3), graded lexical entailment (section 4.4) and the hierarchical path completion (section 4.5).
> Each task has different inference methods, and that is the reason why we describe the inference methods under each section separately and not in the method for learning hierarchical word embeddings section.
> For example, for the first task (section 4.3) we used the concatenation approach as stated in the section.
> Similarly, for the graded lexical entailment task, we used the inference method described in section 4.4 (Eq. (6)).
> The inference methods described in Table 3 are specific to the hierarchical path completion task.
> Among the different inference methods compared in Table 3, ADD corresponds closely to the training objective used by the HWE learning method we propose (see Eq. (2)). This might explain why ADD turns out to be the best inference methods in Table 3.
>
>
> -->Q3: I would also have liked to see a clear motivation for why the GloVE embedding is compatible with and beneficial for the hypernym task. “Relatedness” is different than “hypernymy.”
>
> Ans: All the datasets in the hypernym identification task are pairwise relation data, and it could be the case that it easier for such distributional methods to pick the hypernymy, where hypernymy tend to occur in similar context.

---

### Author Response · Authors · 2019-02-01
**Summary of changes in the new version of the paper.**

We thank all the reviewers for their valuable comments and constructive suggestions. The main concerns highlighted from the reviewers are about the evaluation part, mainly about missing an empirical comparison with some recent relevant work. We have now updated the original paper with:
(1) More related work and discussion about their relevance to the proposed method
(2) An empirical comparison with the suggested relevant work in all the evaluation tasks
(3) A new evaluation task (section 4.4) on the graded lexical entailment.

---

### Meta-Review · Area_Chair1 · 2019-02-10
**Paper with initial unawareness of important related work but convincing revision**

**Recommendation:** Accept (Poster)
**Confidence:** 4

**Metareview:**

All reviewers voiced concerns regarding the comparison to recent related work. However, in my view, the authors addressed these concerns well in their revision, comparing directly against Poincaré embeddings and LEAR. While the comparison reveals mixed results with respect to LEAR, I believe this work is well executed and of interest to the AKBC community.

---

### Decision · Program_Chairs · 2019-02-15
**AKBC 2019 Conference Decision**

Accept